# Lead Chalcogenide Colloidal Quantum Dots for Infrared Photodetectors

**DOI:** 10.3390/ma16175790

**Published:** 2023-08-24

**Authors:** Xue Zhao, Haifei Ma, Hongxing Cai, Zhipeng Wei, Ying Bi, Xin Tang, Tianling Qin

**Affiliations:** 1School of Optics and Photonics, Beijing Institute of Technology, Beijing 100081, China; 3220210544@bit.edu.cn (X.Z.); 3220220493@bit.edu.cn (H.M.); xintang@bit.edu.cn (X.T.); 2Physics Department, Changchun University of Science and Technology, Changchun 130022, China; caihx@cust.edu.cn (H.C.); weizp@cust.edu.cn (Z.W.); 3Beijing Institute of Aerospace Systems Engineering, Beijing 100076, China; centres@foxmail.com

**Keywords:** colloidal quantum dots, infrared photodetector, lead chalcogenide, mercury chalcogenide

## Abstract

Infrared detection technology plays an important role in remote sensing, imaging, monitoring, and other fields. So far, most infrared photodetectors are based on InGaAs and HgCdTe materials, which are limited by high fabrication costs, complex production processes, and poor compatibility with silicon-based readout integrated circuits. This hinders the wider application of infrared detection technology. Therefore, reducing the cost of high-performance photodetectors is a research focus. Colloidal quantum dot photodetectors have the advantages of solution processing, low cost, and good compatibility with silicon-based substrates. In this paper, we summarize the recent development of infrared photodetectors based on mainstream lead chalcogenide colloidal quantum dots.

## 1. Introduction

Infrared photoelectric technologies use infrared materials to convert infrared radiation energy into electrical signals. Infrared photodetectors are widely used in autonomous driving, machine vision, security surveillance, and other fields [1,2,3,4,5]. In recent years, with the continuous development of infrared detection technology, more and more infrared photodetectors have been developed. Currently, the mainstream infrared photodetectors are mainly based on narrow-gap semiconductors such as InGaAs [6,7], HgCdTe [8,9,10], and other materials, which have high sensitivity, wide-band detection capabilities, long-time stability, and good reliability [11]. However, they still have problems such as high cost, a complex epitaxial growth process, and poor compatibility with silicon-based readout integrated circuit chips [12,13,14], limiting their wider application.

The emergence of colloidal quantum dots (CQDs) provides a feasible way for the development of infrared photodetectors. As a new type of photoelectric material, CQDs have the advantages of a wide spectral tuning range, low cost of preparation by thermal injection, and solution processing [15,16,17], and CQDs have great application potential in many practical applications. In addition, CQDs can be directly integrated with silicon-based readout integrated circuits through solution processing, which reduces the fabrication cost and difficulty [18,19]. As a result, CQD infrared detectors have been a research hotspot in recent years. Among CQD infrared detectors [20], lead-based [21] CQD photodetectors have been demonstrated to have high sensitivity and response speed. Lead-based CQDs include PbTe, PbSe, and PbS CQDs. Their detection bands are mainly in the near-infrared and visible light range. Among them, PbSe and PbS CQDs have already been applied to large-array readout integrated circuits [22]. However, lead-based CQDs have the problems of easy oxidation and interference from humidity in the air [23].

In this paper, we discuss the progress of lead chalcogenide colloidal quantum dots for infrared photodetectors. First, we introduce the CQD synthesis method and illustrate the passivation effect of different ligand exchanges on the surface of CQDs. In addition, the development of the device structure of CQD photodetectors is also introduced in detail, and the advancements of photodetectors combining lead chalcogenide CQDs with organic materials, two-dimensional materials, and other materials are discussed.

## 2. PbTe CQD-Based Photodetectors

### 2.1. Synthesis of PbTe CQDs

Lead-based CQDs have received significant attention in recent years due to their unique optical and electronic properties. PbX (Te, Se, S) possess a cubic crystal lattice structure [24,25,26] and are narrow-band-gap semiconductors with 0.32 eV, 0.28 eV, and 0.41 eV, respectively. As a result, PbX (Te, Se, S) CQDs exhibit size-tunable infrared band gaps from visible to infrared. Benefitting from mature chemical synthesis technology, they are attractive for a wide range of applications, including infrared photodetectors, photovoltaics, and thermoelectric devices. PbTe CQDs, in particular, possess a large exciton Bohr radius (46 nm) [27], a high dielectric constant (1000) [28], and a high multiexciton generation yield.

In 1995, Reynoso et al. first grew PbTe CQDs in doped glass, and the absorption wavelength could be adjusted at 1.1–2.0 μm by changing the heat time and temperature [29]. The application potential of PbTe CQDs in optoelectronic devices was demonstrated. In 2006, Murphy et al. reported a synthesis method for spherical and cubic PbTe CQDs [30], whose size distribution could reach 7%, and the diameter of synthesized CQDs ranged from 2.6 to 18 nm. The first exciton transition was achieved from 1009 to 2054 nm (Figure 1a). The photoluminescence quantum yield of spherical PbTe CQDs was as high as 52 ± 2%. In the same year, Urban et al. synthesized monodispersed PbTe CQDs [31] with obvious advantages in size tunability and solution processability compared to the early CQDs in glass (Figure 1b). In addition, the conductivity of the PbTe CQDs film could be increased by 9–10 orders of magnitude through chemical treatment. Therefore, PbTe CQDs were expected to be applied in photodetectors.

Aiming at the problem of PbTe CQDs’ susceptibility to oxidation, in 2009, Lambert et al. proposed a core–shell structure of PbTe/CdTe [32]. The CdTe shell was grown around PbTe using the cation exchange method (Figure 1c). It was observed that PbTe and CdTe had the same lattice orientation, achieving seamless matching. This method effectively resolved the oxidation problem of PbTe CQDs, but it had an anisotropy problem in the exchange process. To improve the method for the synthesis of PbTe CQDs, in 2012, Pan et al. proposed a synthesis method for monodisperse hydrophobic PbTe CQDs [27]. Using oleylamine as the capping ligand and solvent allowed the hydrophobic PbTe CQDs to be easily transformed into CQDs with different ligands. The application of PbTe CQDs in biomedicine was possible. PbTe CQDs could be changed from hydrophobic to hydrophilic through ligand exchange with 4-mercaptopyridine. In addition, the synthesized PbTe CQDs were found to be air-stable.

To enhance the photoluminescence (PL) performance of PbTe CQDs, Protesescu et al. reported the PL properties of core–shell PbTe/CdTe CQDs in 2016 [33]. PbTe/CdTe CQDs were demonstrated to have stable near-infrared emission within the 1–3 μm range. During the cation exchange process, with the addition of excess cadmium oleate, Cd could replace Pb to shrink the PbTe core, resulting in a shift of the PL peak to a shorter wavelength (Figure 1d). Compared with pure PbTe CQDs, the PL stability of PbTe/CdTe CQDs was significantly improved, even in ambient air.

In 2019, Peters et al. studied the fundamental chemistry of PbTe CQDs [34]. The relationship between the band gap and the NC diameter was measured, and the results showed that the energy absorbed by the primary optical absorption had a 1/d relationship with the diameter of the CQDs. And the connection mode of surface ligand oleic acid and PbTe CQDs was mostly chelating bidentate coordination. In 2020, Miranti et al. proposed a core–shell structure of PbTe/PbS CQDs, which established a type II heterojunction that enabled PbTe/PbS CQDs to carry out electron transport exclusively [35] (Figure 1f). The structure exhibited maximum electron mobility of 0.62 cm^2^/Vs with an n-channel current modulation ratio of 104, which was significantly higher than that of PbTe CQDs and PbS CQDs. Enhancing n-type transport made core–shell PbTe/PbS promising for applications in the thermoelectric and electron transport layers of photovoltaic devices.

PbTe CQDs are highly sensitive to oxygen. Modifying the surface of PbTe CQDs or using other CQDs to wrap PbTe CQDs could be considered in future development to prevent the interference of external oxygen. And through the combination of other CQDs and PbTe CQDs, the expansion of the detection band and performance of PbTe CQDs was expected.

**Figure 1 materials-16-05790-f001:**
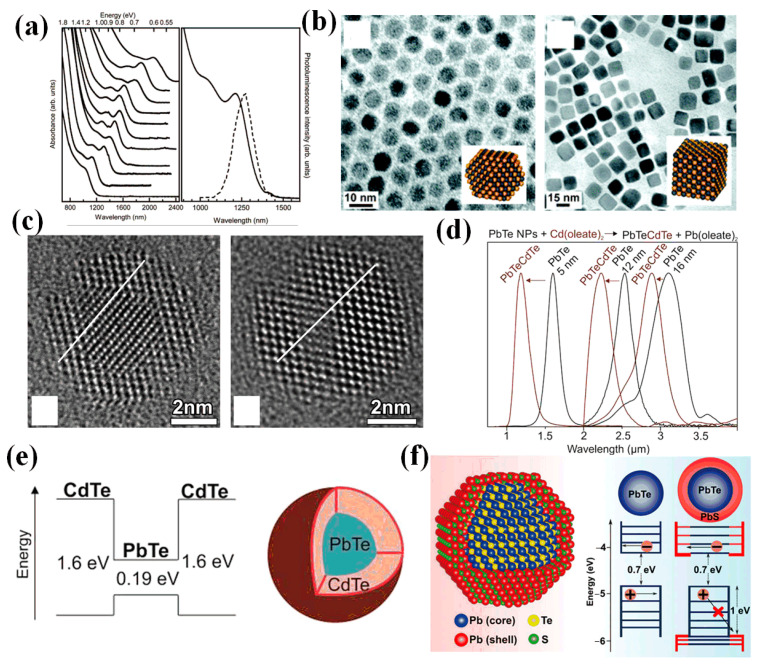
(**a**) The left inset shows the infrared absorption spectra of spherical PbTe CQDs with different sizes, and the right inset shows the absorption (solid line) and photoluminescence (dotted line) spectra of PbTe CQDs with a size of 2.9 nm [30]. Copyright 2006, Journal of the American Chemical Society. (**b**) Transmission electron microscopy (TEM) images and model diagrams of cuboctahedron and cubic PbTe CQDs [31]. Copyright 2006, Journal of the American Chemical Society. (**c**) TEM image of PbTe/CdTe CQDs in the <110> crystal plane [32]. Copyright 2009, Chemistry of Materials. (**d**) PL spectrum of PbTe/CdTe CQDs. (**e**) The energy band diagram of PbTe/CdTe CQDs [33]. Copyright 2016, ChemPhysChem. (**f**) The model and energy band diagram of PbTe/PbS CQDs [35]. Copyright 2020, ACS Nano.

### 2.2. PbTe CQD Photodetectors

The exceptional electronic and optical properties, along with the controllable synthesis, made PbTe CQDs a promising candidate for a wide range of applications in various devices. By acquiring a deeper understanding of the coordination chemistry between oleic acid and PbTe CQDs, we can further optimize the synthesis and properties of PbTe CQDs. This endeavor will lay a robust foundation for their development in future electronic and optoelectronic devices.

The weak binding energy of PbTe CQDs, easy oxidation in air, and low photoconductivity limited their application in photodetectors. In 2016, Lin et al. proposed a PbTe CQD photodetector treated with tetrabutylammonium iodine (TBAI) [36] (Figure 2a,b). The TABI ligand exchange effectively passivated the surface of PbTe CQDs, reducing the average spacing between CQDs from 4.2 nm to 2.3 nm. As a result, the TABI-treated PbTe CQDs film transformed from a nonlinear Schottky contact to a symmetrical Ohmic contact. The maximum responsivity of the device could reach 1.9 mA/W. Under 150 Hz laser irradiation, the response time of the device reached 0.39 ms and the recovery time reached 0.49 ms, which effectively improved the response speed of the device (Figure 2c,d). Moreover, the detector with TBAI-treated PbTe CQDs had a stable photoresponse in air due to the protective effect of the upper polymethyl methacrylate (PMMA) layer.

In 2022, Kate et al. proposed a device structure for selectively growing PbTe CQDs on InP and characterized the CQD devices in both zero and finite magnetic fields (Figure 2e,f) [37]. The Coulomb charge stability analysis revealed large even–odd spacing at zero magnetic field. Further study of the devices under a limited magnetic field showed that PbTe CQDs exhibit a strong Rashba spin–orbit interaction.

The application of PbTe CQD photodetectors had been limited due to the issue of easy oxidation. To overcome this challenge, the investigation of surface ligand exchange for PbTe CQDs could be considered to protect CQDs from oxidation and enhance their stability. Alongside ligand exchange, exploring other surface passivation strategies was also important. This included the utilization of inorganic passivation materials such as metal oxides and chalcogenides, as well as the development of hybrid passivation methods combining organic and inorganic materials. The study of surface passivation strategies for PbTe CQDs plays a crucial role in improving their stability and enabling their practical application in optoelectronic devices.

## 3. PbSe CQD-Based Photodetectors

### 3.1. Synthesis of PbSe CQDs

PbSe CQDs are a material with strong quantum confinement, having an exciton Bohr radius of 46 nm [38]. This characteristic contributes to stronger electron coupling, which leads to improved charge carrier transport within the PbSe CQD film.

In 1997, Lipovskii et al. successfully synthesized size-controllable PbSe CQDs with a size between 2 and 15 nm and a size distribution of about 7% [39]. In 2004, Pietryga et al. studied the mid-infrared PL properties of PbSe CQDs [40]. The experimental results showed that the synthesized PbSe CQDs had excellent fluorescence tunability and narrow size distribution (Figure 3a,b). In 2007, Baek et al. synthesized monodisperse PbSe CQDs using solution chemistry and realized the use of two different ligands of acetic acid (AA) and hexanoic acid (HA) to control the size of CQDs [41] (Figure 3c). The experimental results showed that AA and HA could replace part of the oleate on the surface of PbSe CQDs, resulting in a reduction in steric hindrance and a reduction in the average size of CQDs. To enhance the antioxidant ability of PbSe CQDs, in 2012, Hughes et al. proposed a method of ligand exchange for PbSe CQDs using alkyl selenides [42] (Figure 3d). Alkyl selenides could completely replace the oleic acid ligands on the surface of CQDs. Compared with the oleic acid ligand, the combination between the Pb atom on the surface of the alkyl selenide ligand and the alkyl selenide ligand was more stable, thereby improving the oxidation resistance of the surface of CQDs.

In 2017, Campos et al. proposed a method for the synthesis of PbSe CQDs through N, N, N′-Trisubstituted selenourea precursors [43]. The size of CQDs (1.7–6.6 nm) could be well regulated, with a narrow size distribution of 0.5–2% and a narrow spectral absorption peak (Figure 3e). Experiments had proved that the N, N, N’-Trisubstituted selenourea had a slower and more controllable conversion reaction than the N, N’-Trisubstituted selenourea and the N, N, N’, N’-tetrame-thylselenourea. The proposal of trisubstituted selenourea provided a feasible strategy for other CQDs synthesized from selenium precursors. PbSe CQDs usually required complex and tedious synthesis steps. In 2020, Liu et al. proposed a method for the direct synthesis of PbSe CQDs at room temperature in one step, which simplified the synthesis steps and costs [44] (Figure 3f). The experimental results confirmed that CQDs had excellent surface passivation ability.

### 3.2. PbSe CQD Photodetectors

Due to the gradual maturity of PbSe CQD synthesis technology, photodetectors based on PbSe CQDs were studied. In 2010, Sarasqueta et al. first proposed an ITO-PbSe-Al infrared photodetector prepared with PbSe CQDs, and the responsivity could reach 0.67 A/W [45] (Figure 4a). The PbSe surface was passivated by chemical treatment, which effectively reduced the dark current of the device. The performance of the devices under different ligand exchanges was also explored, and the current density of the device treated with benzonedthiol (BDT) was 2 orders of magnitude higher than that of the untreated device. Treatment with BDT showed a 20-fold increase in hole mobility and an 80-fold increase in electron mobility. In 2016, Fu et al. enhanced the stability of synthesized PbSe CQDs by ammonium chloride treatment and prepared Au/PbSe/PMMA/Au photodetectors [46] (Figure 4b). The responsivity of the device reached 64.17 mA/W, and the detectivity reached 5.08 × 10^10^ Jones at 980 nm. This work on PbSe CQDs demonstrated a highly promising approach for the development of infrared detectors.

To extend the detection band of PbSe CQD photodetectors, in 2019, Zhu et al. proposed a PbSe CQD photodetector with a detection range from the ultraviolet to the mid-infrared region (350–2500 nm) [47] (Figure 4c,d). The hole-trapping-induced photomultiplication effect was realized by p-type EDT (ethanedithiol)-PbSe CQDs and an n-type TABI-PbSe CQDs double-layer film. Experimental results showed that the photodetector had a quantum efficiency of 450% in the visible region and a detectivity of more than 10^12^ Jones at room temperature. In the infrared region, the quantum efficiency could reach 120% and the detectivity could reach 4 × 10^11^ Jones. Such photodetectors provided a feasible way for the development of uncooled, broadband photodetectors.

In order to achieve PbSe CQD compatibility with silicon-based substrates, in 2021, Chen et al. proposed a photodetector using PbSe CQDs in Si, with an infrared spectral response range of 450–1550 nm [48] (Figure 4e,f). The device achieved a responsivity of 648.7 A/W and a response time of 32.3 μs at 1550 nm. The detectivity reached 7.48 × 10^10^ Jones, and the external quantum efficiency reached 6.47 × 10^4^%. The device had the advantages of being low-cost and easy to integrate with silicon-based readout circuits. In the same year, Peng et al. proposed a near-infrared detector using PbSe CQDs [49] (Figure 4g,h). PbSe CQDs were created using the method of one-step synthesis at room temperature and directly introduced iodide ions, which realized in situ passivation of iodide ions and avoided redundant ligand exchange processes. The device had a responsivity of 970 mA/W and a detectivity of 1.86 × 10^11^ Jones at 808 nm. Compared with the pure PbSe CQD detector, the photocurrent of the heterojunction detector increased from 7.6 × 10^−9^ A to 7.4 × 10^−8^ A, and the dark current decreased from 1.3 × 10^−9^ A to 7.7 × 10^−11^ A, effectively reducing the dark current of the device.

Since pure PbSe CQDs were easily oxidized, which led to the degradation of the device performance, the concept of combining PbSe CQDs with other materials had been proposed. A composite material was prepared with materials such as organic polymers and metal oxides, enabling the interaction between different materials to optimize device performance. It provided a novel way to achieve an extended band and improve device stability.

In 2015, Wang et al. proposed a near-infrared photodetector with a field-effect transistor structure combining PbSe CQDs and poly (3-hexylthiophene-2, 5-diyl) (P3HT) [50] (Figure 5a). Responsivity and detectivity reached 500 A/W and 5.05 × 10^12^ Jones, respectively. Two-dimensional materials became a research hotspot because of their excellent optical and electrical properties. Two-dimensional materials have high carrier performance, so combining with CQDs could make up for the shortcomings of CQDs. In this combination, a quantum dot layer is used as a photosensitive material, and a two-dimensional material can improve the mobility of carriers and other properties. Therefore, devices combining zero-dimensional and two-dimensional materials could effectively improve the performance of detectors. In 2019, Luo et al. reported a device combining PbSe CQDs and Bi_2_O_2_Se [51] (Figure 5b,c), realizing 2 μm short-wave detection. The responsivity was greater than 103 A/W, and the response time could reach 4 ms. High sensitivity and quick response were realized. In 2022, Peng et al. reported a photodetector integrated with PbSe CQDs and two-dimensional material MoS_2_ [52] (Figure 5j). The photo–dark current ratio of the heterogeneous photodetector could reach 10^2^, the maximum responsivity could reach 23.5 A/W, and the maximum detectivity could reach 3.17 × 10^10^ Jones under 635 nm illumination. And the detection band of the detector could be extended to the near-infrared region. Under the illumination of 808 nm, it achieved a responsivity of 19.7 A/W and a detectivity of 2.65 × 10^10^ Jones. The proposed heterojunction photodetector improved the performance of photodetectors based on zero- and two-dimensional materials.

Due to the rise of perovskite materials, in 2021, Hu et al. proposed a flexible broadband photodetector based on a CsPbBr_3_/PbSe CQD heterostructure, which took advantage of the excellent properties of perovskite nanocrystalline materials and detected a wide range of wavelengths, from ultraviolet to long-wave infrared [53] (Figure 5e,f). The responsivity of the device was 7.17 A/W and the detectivity was 8.97 × 10^12^ Jones under 365 nm light and 5 V voltage. Response rise and decay times were 0.5 ms and 0.78 ms, respectively. Moreover, the flexible detector could retain 91.2% of its initial performance even after being bent thousands of times. It provided a feasible strategy for the development of flexible devices. In 2022, Sulaman et al. proposed a self-powered broadband photodetector combining PbSe CQDs with CsPbBr_1.5_I_1.5_ [54] (Figure 5g–i). The responsivity could reach 6.16 A/W, and the detectivity could reach 5.96 × 10^13^ Jones.

Due to the excellent electrical and optical properties of PbSe CQDs, in 2021, Dortaj et al. suggested a 10 × 10-pixel high-speed mid-infrared (3–5 μm) camera based on PbSe/PbI_2_ core–shell CQDs [55] (Figure 5k–m). The procedure involved spin-coating PbSe/PbI_2_ CQDs onto the substrate of interdigitated electrodes, followed by surface passivation with epoxy resin. Subsequently, the imaging results were obtained on the display through the readout circuit, as shown in Figure 5l,m. Experimental results showed that the response rise time of the detector could reach 100 ns. The camera could achieve 1 million frames per second, so high-resolution images were obtained.

As a new type of photoelectric conversion device, PbSe CQD photodetectors offer several advantages, including fast response, high sensitivity, low cost, and easy preparation. The performance of PbSe CQDs could be further improved by combining them with other materials. To further improve performance, one approach is to employ a new ligand solution for surface modification of PbSe CQDs. Additionally, the integration of new materials or optical devices can also be considered to achieve performance enhancements.

## 4. PbS CQD-Based Photodetectors

### 4.1. Synthesis of PbS CQDs

PbS CQDs have a wide tunable band-gap range (0.6–1.6 eV), a high molar broadband absorption coefficient (≈10^6^ M^−1^ cm^−1^), and a large Bohr exciton radius (~18 nm) [27]. These characteristics make PbS CQDs excellent candidates for low-cost broad-spectrum photodetectors.

In 1990, Nenadovic et al. first prepared 4 nm PbS CQDs in an aqueous solution [56]. In 2011, Lingley et al. proposed a new method of ligand exchange by replacing the oleic acid ligands on the surface of PbS CQDs with nonanoic and dodecanoic acid ligands [57] (Figure 6a). And a high quantum efficiency (55%) could be maintained. In 2013, Zhang et al. synthesized PbS CQDs with good size distribution using H_2_S as a sulfur source, and the stability and reproducibility of CQDs prepared by this method were better [58] (Figure 6b). In addition, the long ligand of oleic acid on the surface of PbS CQDs was replaced by a short ligand (butylamine), which could effectively extend the carrier lifetime. In 2017, Lin et al. proposed a liquid-phase ligand exchange method for PbS CQDs, transferring PbS CQDs to the polar solvent DMF for ligand exchange [59] (Figure 6c). The ligand-exchanged PbS CQDs could achieve enhanced mobility and were stable for several months, which laid a solid foundation for the development of photodetectors in the future. In the same year, to produce stable PbS CQDs, Shestha et al. proposed a method of pre-combining thiol ligands with Pb^2+^ before ligand exchange [60], which effectively improved the ligand exchange efficiency of PbS CQDs (Figure 6d). Experimental results showed that 78% of the original oleic acid-terminated CQD photoluminescence quantum efficiency could be maintained after ligand exchange in PbS CQDs by Pb-thiolate. To further study the influence of reaction conditions on PbS CQDs, in 2020, Shuklov et al. proposed a method to synthesize 1.7–2.05 μm PbS CQDs through a mixture of oleic acid and oleylamine [61]. The experimental study showed that the reaction temperature could significantly affect the size of PbS CQDs, while the reaction time had little effect on the PbS CQDs. Good monodisperse PbS CQDs could be obtained when Pb: S = 3: 1. At present, most of the syntheses of PbS CQDs use PbO as the precursor of lead, but it increases the hydroxyl ligands on the surface of PbS CQDs. In 2023, Wang et al. used lead (II) acetylacetonate as the lead precursor to reduce the influence of the hydroxyl ligands on the surface of PbS CQDs [62], and the synthesized PbS CQDs could achieve better binding to iodine ligands during the ligand exchange (Figure 6e). Thus, the surface passivation of PbS CQDs was improved, and the carrier transport ability was enhanced.

### 4.2. PbS CQD Photodetectors

PbS CQDs are widely used in photodetectors because of their simple preparation, spectral tunability, and wide wavelength response. In 2006, Konstantatos et al. reported an infrared photodetector based on PbS CQDs [63] (Figure 7a,b). The normalized detectivity of the device at 1.3 μm at room temperature reached 1.8 × 10^13^ Jones, and the responsivity reached 10^3^ A/W. Its performance was comparable to that of epitaxially grown InGaAs detectors. In 2014, Liu et al. first reported a flexible NO_2_ gas sensor based on PbS CQDs [64] (Figure 7c,d), which achieved high sensitivity and reproducible performance at room temperature. The experimental results showed that p-type doping was achieved by adding NO_2_ on the surface of PbS CQDs. The smaller binding energy was favorable for the rapid desorption of NO_2_ from the surface of PbS CQDs, which led to the rapid recovery of the signal. In order to promote the development of photoconductive devices, in 2016, Iacovo et al. proposed a photoconductive device based on PbS CQDs [65] (Figure 7e). The device achieved high responsivity and detectivity at 1.3 μm under 1 V bias, and the maximum values could reach 30 A/W and 2 × 10^10^ Jones. It was expected to realize applications in silicon integrated circuits.

Due to the low performance of single-layer CQDs, in 2017, Ren et al. proposed a photodetector combining PbS-TABI and PbS-EDT [66] (Figure 7f–h). The detectivity of the double-layer CQD device could reach 1.71 × 10^12^ Jones under 580 nm illumination, which was about 3 times higher than that of the single-layer CQD device. The photo–dark current ratio of the bilayer device was 152.35, much larger than that of PbS-EDT (13.41) and PbS-TABI devices (36.9). In the same year, Qiao et al. further improved the double-layer CQD device and proposed the photodetector structure of PbI_2_/PbS-PbI_2_/PbS-EDT [67] (Figure 7i). The detector could increase the photocurrent while reducing the dark current and achieved a detectivity of 1.3 × 10^13^ Jones and a responsivity of 0.43 A/W. Since silicon-based photodetectors were gradually applied to the field of photodetection, in 2020, Xu et al. proposed a photodetector combining n-type silicon with p-type PbS CQDs [68] (Figure 7j,k). The device had a lower band offset and better charge trapping. Under 1540 nm illumination, a detectivity of 1.47 × 10^11^ Jones and a responsivity of 0.264 A/W were achieved. In the same year, Shi et al. proposed a silicon-compatible PbS CQD photodetector that could be integrated into a chip [69] (Figure 7l). The detectivity reached 3.95 × 10^12^ Jones, and the external quantum efficiency could reach 4.96 × 10^5^%. This photodetector could realize a broadband response wavelength in the 405–1550 nm range. The mature silicon processing technology and rational band optimization of CQDs further enhanced the capabilities of PbS CQD-based photodetectors.

Detectors based on pure PbS CQDs were still limited in the detection band, so the purpose of extending the band and improving performance was achieved by combining them with other materials. In 2009, Krisztina et al. proposed a photodetector combined with [6,6]-phenyl-C61-butyric acid methyl ester (PCBM) and PbS CQDs (Figure 8a), and its spectral range covered the visible and near-infrared regions [70]. The detectivity obtained at 1200 nm was 2.5 × 10^10^ Jones. In 2014, He et al. reported a flexible photodetector combined with Ag nanocrystals (NCs) and PbS CQDs (Figure 8b–e), which could effectively reduce dark current and enhance photocurrent [71]. Experiments proved that adding 0.5% to 1% Ag NCs could effectively improve the detection ability of the device. Ag NCs could capture photogenerated electrons, effectively prolonging the lifetime of carriers, thereby improving the photocurrent. The detectivity of the flexible device could reach 1.7 × 10^10^ Jones. The combination of Ag NCs and CQDs opened up a new strategy for future detector development.

To realize low-cost and high-performance optoelectronic devices, in 2017, Bessonov et al. proposed a method of coupling PbS CQDs with CH_3_NH_3_PbI_3_ (Figure 8f,g), and the detectivity could reach 5 × 10^12^ Jones [72]. The addition of CH_3_NH_3_PbI_3_ enabled PbS CQDs to generate greater gains. To further reduce the dark current of the device, in 2020, Ka et al. reported a strategy combining copper thiocyanate (CuSCN) and PbS CQDs to reduce the dark current of PbS CQD-based photodetectors [73] (Figure 8h,i). The detectivity of the prepared photodiode could reach 10^11^ Jones. The dark current was reduced by 2 orders of magnitude, effectively reducing the dark current of the photoelectric device.

Since PbS CQDs could realize photodetection in the visible to infrared region, and CQDs had the advantages of solution processability and substrate compatibility, PbS CQDs were applied in readout circuits. In 2015, Klem et al. proposed a short-wavelength readout circuit photodiode based on PbS CQDs, which consisted of a linear 320-pixel array [74]. The dark current density at room temperature was 6.8 nA/cm^2^, and the specific detectivity could reach 10^12^–10^13^ Jones, which was comparable to that of the InGaAs photodiode.

Due to the rise of graphene materials, in 2017, Goossens et al. proposed a 388 × 288 array complementary metal oxide semiconductor (CMOS) imaging system combining PbS CQDs and graphene [75] (Figure 9a,b). The CQD layer and the graphene layer formed a vertical heterojunction; the graphene layer trapped holes, and the CQD layer trapped electrons. The detectivity reached up to 1012 Jones in the range of 300–2000 nm.

In 2019, Zhang et al. further optimized a device combining PbS CQDs with CH_3_NH_3_PbI_3_ and fabricated a 10 × 10 array photodetector [76] (Figure 9c). It realized a wide-band detection capability in the ultraviolet–visible–near-infrared region, the responsivities could reach 255 A/W and 1.58 A/W at 365 nm and 940 nm, respectively, the detectivities at 365 nm and 940 nm were 4.9 × 10^13^ Jones and 3.0 × 10^11^ Jones, and the response time was 42 ms. The appearance of this detector laid the foundation for the development of high-sensitivity broadband photodetectors and imagers. In the same year, Georgitzikis et al. proposed an imager combining a polymer (organic) with PbS CQDs [78]. The photodiode combined with the polymer and PbS CQDs was integrated into a 512 × 768-pixel focal plane array to realize imaging in visible and infrared environments. The maximum detectivity could reach 2.3 × 10^12^ Jones, the minimum rise response time was 13 μs, and the minimum fall response time was 30 μs. High-resolution and high-sensitivity infrared imaging was realized. In 2020, Choi et al. proposed a 1 × 6 linear array photodetector combined with PbS CQDs and InGaZnO [77] (Figure 9d). At 1310 nm, the detector achieved a responsivity of 10^4^ A/W and a detectivity of 10^12^ Jones. Imaging of the pattern was finally achieved (Figure 9e). The surface modification of PbS CQDs with TABI increased the stability and oxidation resistance of the devices. However, the devices surface-modified with EDT lost their detection ability within 2 weeks. In 2022, Liu et al. proposed a 640 × 512-pixel high-resolution imager based on PbS CQDs, with a spectral range of 400–1300 nm [21] and a detectivity of 2.1 × 10^12^ Jones, and the spatial resolution of the imager could reach 40 line-pairs per millimeter (Figure 9f,g). Experiments showed that the imager realized vein imaging and substance identification, which promoted the development of PbS CQDs. The PbS CQD imager had the advantages of high sensitivity, high resolution, fast response, and multispectral imaging.

Consequently, compared with single-photon detectors, array detectors have broad development prospects. But in future development, further cost reduction and smaller size are required. Meanwhile, during the synthesis process, PbS CQDs can be susceptible to oxidation and volatility, necessitating the implementation of additional protective measures in future developments.

In addition to the aforementioned CQD detectors, lead chalcogenide photodetectors also encompass bulk semiconductor photodetectors. However, lead chalcogenide CQD photodetectors have numerous advantages over lead chalcogenide bulk photodetectors.

(1)The band structure of lead chalcogenide bulk materials is relatively fixed and difficult to control, resulting in a limited spectral tuning range. In contrast, the band structure of lead chalcogenide CQDs can be adjusted by tuning their size, thereby expanding the spectral tuning range and leading to broader potential applications.(2)Lead chalcogenide bulk semiconductor thin films are usually prepared using the chemical bath deposition method [79,80], which poses challenges in integrating bulk materials with silicon-based readout circuits. In contrast, CQDs are synthesized using a thermal injection method, leading to lower manufacturing costs for CQD detectors. Moreover, CQDs can be directly integrated with silicon-based readout circuits through solution processing, thereby expanding the potential applications of lead chalcogenide CQD photodetectors.(3)Photodetectors based on lead chalcogenide bulk materials need to undergo high-temperature sensitization at 300–600 °C in a specific atmosphere, such as an oxygen-rich and iodine-rich atmosphere [81]. However, the existing sensitization process lacks repeatability, stability, and uniformity, thereby restricting their application [82,83]. Lead chalcogenide CQD photodetectors can operate at room temperature, reducing the manufacturing cost and difficulty.(4)Lead chalcogenide CQD photodetectors can be self-assembled in vertical or horizontal directions, forming more complex structures. The feature provides lead chalcogenide CQD photodetectors with a distinct advantage in terms of integration and multi-channel detection.

## 5. Conclusions

In this review, we reviewed the development of lead chalcogenide CQD photodetectors in recent years. Table 1 summarizes the performance of lead-based CQD photodetectors. The lead-based CQDs include PbTe, PbSe, and PbS CQDs. One can see that lead chalcogenide CQD photodetectors have developed quickly, developing from single-pixel photodetectors to large array imagers. However, compared with traditional bulk semiconductor photodetectors, CQD-based photodetectors still face challenges, such as low detectivity and large dark current. There are several challenges that need to be addressed:

(1)CQD surface passivation. CQD have large surface to volume ratio. As a result, they are overly sensitive to the environment. Surface modification with organic or inorganic ligands could improve CQD stability and protect their physical properties. For instance, PbTe CQDs, are susceptible to oxidation, making them less suitable for photodetector applications. To address the problem of oxidation, surface modification techniques utilizing organic or inorganic ligands can be employed to enhance the stability and photoelectric conversion efficiency of PbTe CQDs. This improvement is expected to enhance the overall photoelectric performance and lifespan of photodetectors. In addition, the directional assembly of CQDs and the fine regulation of their optical properties can be achieved through surface modification.(2)Dark current reduction on CQD-based photodetectors. Compared with InGaAs and HgCdTe based photodetectors, CQD-based photodetectors typically suffer the disadvantage on large dark current. The dark current is usually generated by the surface defects on the CQDs, which can trap and recombine charges. Additionally, thermal excitation in CQD-based devices can lead to dark current generation. To solve this problem, reducing the surface defects and band tail regulation should be the key. In addition, transport property improvement would also be useful such as doping density and mobility modification.(3)Large array photodetectors. At present, most research focus on single-pixel CQD detectors. However, in real application, it is usually necessary to use array detectors. Large area array photodetectors can be prepared by nanoimprinting technology and micro-nano processing technology. There are many technical challenges need to be solved.(4)Broad band photodetectors. At present, the main research on lead based CQD photodetector could only cover near-infrared to short-wave infrared. More research is necessary to promote the progress on broad band photodetection. For example, combining PbS CQDs with graphene, perovskite, and other materials can achieve detection in the visible and near-infrared bands. Therefore, combining CQDs with other materials achieves the purpose of broad-spectrum detection.

## Figures and Tables

**Figure 2 materials-16-05790-f002:**
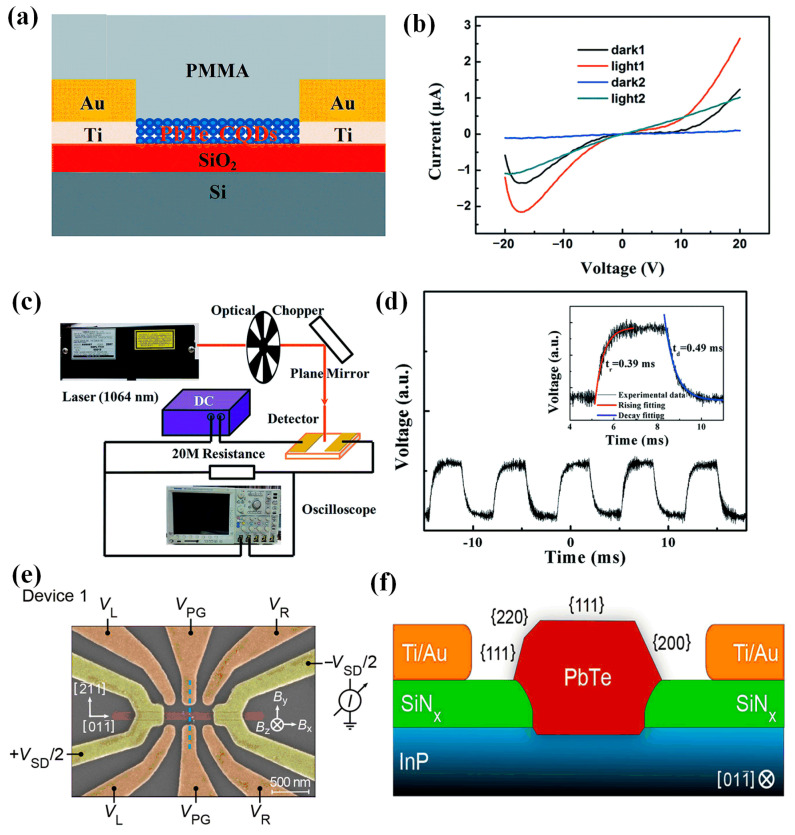
(**a**) Schematic diagram of the structure of the PbTe photodetector. (**b**) I–V curve of the PbTe photodetector. (**c**) Schematic diagram of the experimental setup for testing the photoresponse of the PbTe photodetector. (**d**) Photoresponse curve of the PbTe photodetector under laser irradiation frequency of 150 Hz [36]. Copyright 2016, RSC Advances. (**e**) Scanning electron microscope (SEM) image of PbTe device with crystal and magnetic field. (**f**) Schematic diagram of PbTe device [37]. Copyright 2022, Nano Letters.

**Figure 3 materials-16-05790-f003:**
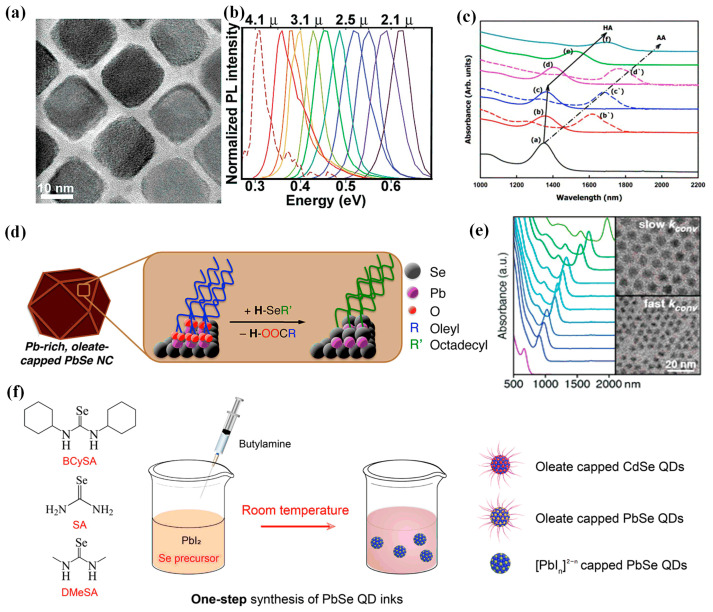
(**a**) TEM image of PbSe CQDs. (**b**) PL spectra of CQDs with different sizes at room temperature (size distribution = 10%) [40]. Copyright 2004, Journal of the American Chemical Society. (**c**) Infrared absorption spectra of PbSe CQDs with different contents of AA and HA. The molar ratios of PbO:Se:oleic acid (OA): HA = 1:3:4.5:X, where X = 0.002, 0.02, 0.1, 0.2, 0.5, 1.0, (a–f). The molar ratio of PbO:Se:OA:AA = 1:3:4.5:X, where X = 0.02, 0.1, 0.2, (b’–d’) [41]. Copyright 2007, Journal of Colloid Interface Science. (**d**) Process for surface ligand exchange of PbSe CQDs using alkyl selenides [42]. Copyright 2012, ACS Nano. (**e**) Absorption spectrum and TEM images of PbSe CQDs synthesized using N, N, N’-Trisubstituted selenourea [43]. Copyright 2017, Journal of the American Chemical Society. (**f**) Procedure for the direct synthesis of PbSe CQDs using a one-step approach [44]. Copyright 2020, ACS Energy Letters.

**Figure 4 materials-16-05790-f004:**
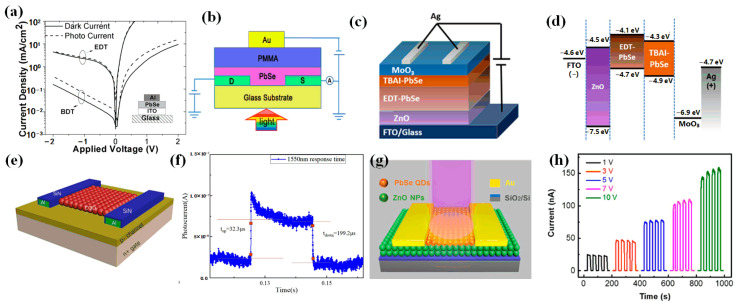
(**a**) I-V curves of the PbSe CQD photodetectors treated with EDT and BDT; the inset is the device schematic of the photodetector [45]. Copyright 2010, Chemistry of Materials. (**b**) Cross-sectional schematic of a FET-based photodetector [46]. Copyright 2016, Nanotechnology. (**c**) Schematic of a device based on bilayer PbSe CQDs. (**d**) Schematic diagram of energy bands based on bilayer PbSe CQDs [47]. Copyright 2019, ACS Applied Materials & Interfaces. (**e**) Schematic diagram of the device of Si: PbSe CQDs. (**f**) Response time curve of Si: PbSe CQDs [48]. Copyright 2021, Journal of materials science. (**g**) Schematic diagram of PbSe CQD photodetector. (**h**) Photoresponse of PbSe CQD photodetector with different voltage biases [49]. Copyright 2021, ACS Applied Materials & Interfaces.

**Figure 5 materials-16-05790-f005:**
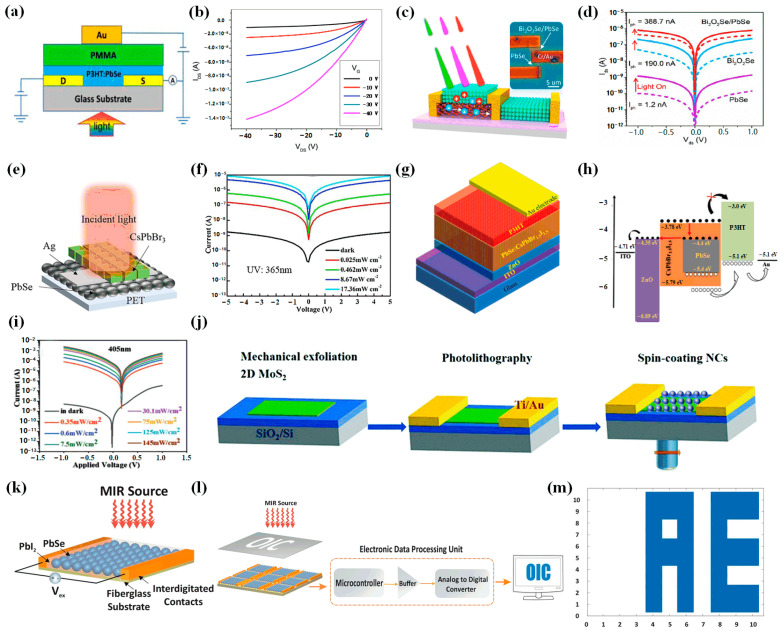
(**a**) Schematic diagram of P3HT: PbSe CQD photodetector. (**b**) Output characteristics of FET-based photodetector Au(Gate)/PMMA/P3HT:PbSe/Au(Source, Drain) in dark [50]. Copyright 2015, IEEE Photonics Technology Letters. (**c**) Schematic illustration of PbSe/Bi_2_O_2_Se photodetector. (**d**) I−V curves of bare Bi_2_O_2_Se, PbSe, and PbSe/Bi_2_O_2_Se hybrid photodetectors under dark and illumination (532 nm, 3.7 mW/cm^2^) [51]. Copyright 2019, ACS Nano. (**e**) Schematic diagram of CsPbBr_3_/PbSe photodetector. (**f**) Typical I–V curves of the CsPbBr_3_/PbSe heterostructure-based PDs under 365 nm [53]. Copyright 2021, Journal of Materials Science & Technology. (**g**) Schematic diagram of PbSe/CsPbBr_1.5_I_1.5_. (**h**) Energy band structure diagram of PbSe/CsPbBr_1.5_I_1.5_. (**i**) I–V curves of photodetector ITO/ZnO/PbSe:CsPbBr_1.5_I_1.5_/P3HT/Au in dark and under 405 nm [54]. Copyright 2022, Advanced Functional Materials. (**j**) Schematic diagram of PbSe/MoS_2_ fabrication process [52]. Copyright 2022, Journal of Materials Chemistry C. (**k**) Schematic diagram of PbSe/PbI_2_ photodetector. (**l**) Schematic diagram of the imaging process of the camera. (**m**) “AE” logos obtained by the camera [55]. Copyright 2021, Scientific Reports.

**Figure 6 materials-16-05790-f006:**
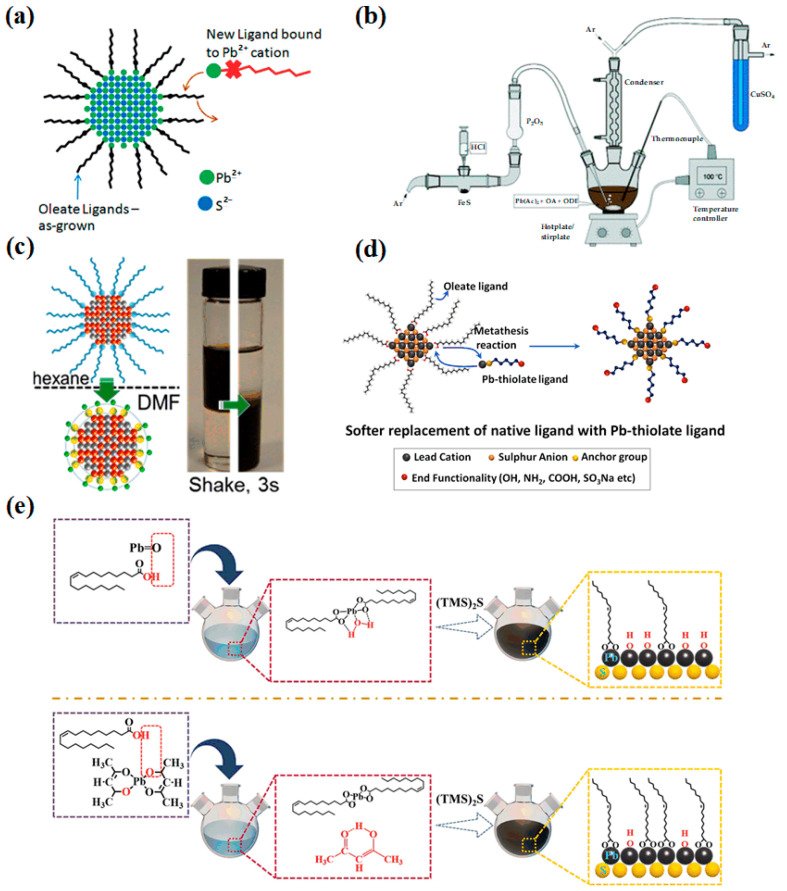
(**a**) Schematic diagram of ligand exchange [57]. Copyright 2011, Nano Letters. (**b**) Diagram of the synthesis setup for PbS CQDs [58]. Copyright 2013, CrystEngComm. (**c**) Schematic diagram of the ligand exchange process. The right inset is the phase transfer photo of PbS CQDs [59]. Copyright 2017, Journal of the American Chemical Society. (**d**) Schematic diagram of the ligand exchange using Pb-thiolate [60]. Copyright 2017, Small. (**e**) Schematic diagram of PbS CQD synthesis process using lead oxide and lead acetylacetonate [62]. Copyright 2023, Advanced Science.

**Figure 7 materials-16-05790-f007:**
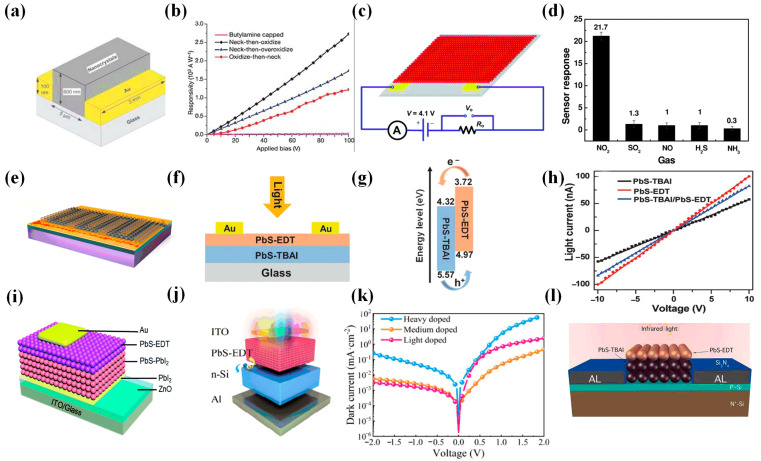
(**a**) Schematic diagram of PbS CQD device. (**b**) Responsivity as a function of applied bias. Necked nanocrystal devices show comparable responsivities, consistent with similar carrier mobilities and trap state lifetimes [63]. Copyright 2006, Nature. (**c**) Schematic illustration of NO_2_ gas sensors. (**d**) Response of PbS CQDs sensors to different gases (50 ppm) [64]. Copyright 2014, Advanced Materials. (**e**) Schematic diagram of PbS CQD photodetector on interdigitated contacts [65]. Copyright 2016, Scientific Reports. (**f**) Schematic diagram of PbS-EDT/PbS-TABI photodetector. (**g**) Band diagram of PbS-EDT/PbS-TABI photodetector. (**h**) I–V curves of the three devices under a white light illumination of 0.20 mW/cm^2^ [66]. Copyright 2017, Advance Materials. (**i**) Schematic diagram of PbS-EDT/PbS-PbI_2_ photodetector [67]. Copyright 2017, RSC Advances. (**j**) Schematic diagram of PbS CQD photodetector. (**k**) Dark current density of Si/PbS PD made using light-doped, medium-doped, and heavy-doped Si wafers [68]. Copyright 2020, ACS Applied Materials & Interfaces. (**l**) Schematic diagram of PbS-EDT/PbS-TABI/Si photodetector [69]. Copyright 2020, Nanotechnology.

**Figure 8 materials-16-05790-f008:**
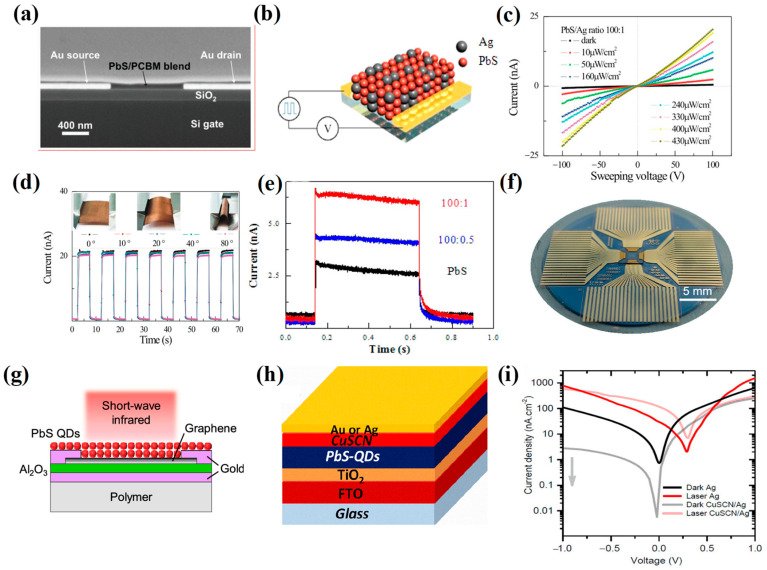
(**a**) SEM of the PbS/PCBM device [70]. Copyright 2009, Advanced Materials. (**b**) Schematic diagram of PbS CQD/Ag NC photodetector. (**c**) I−V characteristics of the 1% Ag NC composite device under different levels of light illumination ranging from 10 to 430 μW/cm^2^. (**d**) Current as a function of time. The inset shows the detector bent at different angles. (**e**) Dynamic current–time (I−T) curves of PbS CQD/Ag NC composite (100:0, 100:0.5, and 100:1) photodetectors under 36 μW/cm^2^ illumination and 40 V voltage [71]. Copyright 2014, ACS Photonics. (**f**) Photograph of PbS/graphene photodetector on a silicon wafer. (**g**) Schematic diagram of PbS/graphene [72]. Copyright 2017, ACS Nano. (**h**) Schematic diagram of PbS/CuSCN photodetector. (**i**) I–V characteristics of the photodiodes fabricated with and without the CuSCN film in the dark and under illumination with a 532 nm laser at 0.5 mW/cm^2^ [73]. Copyright 2020, ACS Photonics.

**Figure 9 materials-16-05790-f009:**
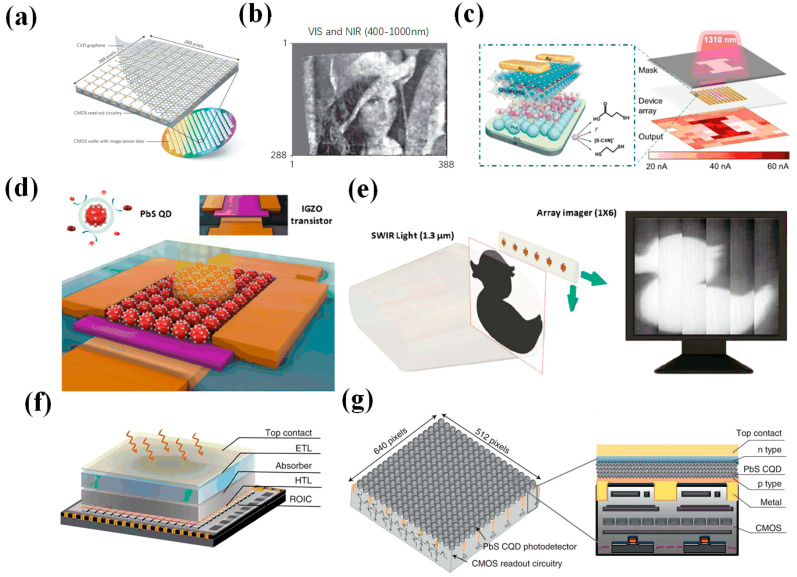
(**a**) Schematic diagram of graphene transfer process on 388 × 288 array image sensor. (**b**) Visible and near-infrared photograph of a standard image reference “Lena” printed in black and white on paper illuminated with an LED desk lamp [75]. Copyright 2017, Nature Photonics. (**c**) Device structure of PbS/CH_3_NH_3_PbI_3_ photodetector. The right inset shows photoelectric imaging of the letter “I” under near-infrared light [76]. Copyright 2019, ACS Applied Materials & Interfaces. (**d**) Schematic diagram of PbS CQD/InGaZnO photodetector. (**e**) Imaging schematic diagram of the PbS CQD/InGaZnO photodetector [77]. Copyright 2020, ACS Photonics. (**f**) Schematic diagram of PbS CQD readout circuit. (**g**) Schematic diagram of PbS CQD imager and cross-section of PbS CQD imager [21]. Copyright 2022, Nature Electronics.

**Table 1 materials-16-05790-t001:** The summary of CQD-based photodetectors.

Year	Photoactive Material	Detection Range (nm)	Detectivity (Jones)	Responsivity (A/W)	Rise Decay Time	Refs.
2006	PbS CQDs	1300	1.8 × 10^13^	10^3^	--	[63]
2009	PbS CQDs/PCBM	1200	2.5 × 10^10^	1.6	--	[70]
2010	PbSe CQDs	1400	--	0.67	--	[45]
2014	PbS CQDs/Ag NCs	1100	1.7 × 10^10^	0.0038	--	[71]
2015	PbSe CQDs/P3HT	980	5.05 × 10^12^	500	--	[50]
2016	PbSe CQDs	980	5.08 × 10^10^	0.06417	--	[46]
2016	PbTe CQDs	1064	--	0.0019	0.39 ms 0.49 ms	[36]
2016	PbS CQDs	1300	2 × 10^10^	30	160 ms 3 s	[65]
2017	PbS-EDT/PbS-TABI	580	1.71 × 10^12^	0.25	3.63 ms 29.56 ms	[66]
2017	PbS-EDT/PbS-PbI_2_	850	10^13^	0.43	5.3 μs 4.9 μs	[67]
2017	PbS CQDs/CH_3_NH_3_PbI_3_	520	5 × 10^12^	2 × 10^5^	10 ms 0.5s	[72]
2019	PbS CQDs/CH_3_NH_3_PbI_3_	365/940	4.9 × 10^13^@365 nm 3.0 × 10^11^@940 nm	255@365 nm 1.58@940 nm	42 ms --	[76]
2019	PbSe-TABI/PbSe-EDT	1300/2400	10^12^@1300 nm 10^11^@2400 nm	0.05–0.2	140 μs 410 μs	[47]
2019	PbSe CQDs/Bi_2_O_2_Se	2000	--	10^3^	4 ms	[51]
2020	PbS-TABI/PbS-EDT	1540	1.47 × 10^11^	0.264	2.04 μs 5.34 μs	[69]
2020	PbS CQDs/CuSCN	532	10^11^	--	50 μs 110 μs	[73]
2021	PbSe CQDs	1550	7.48 × 10^10^	648.7	32.3 μs 73.2 μs	[48]
2021	PbSe CQDs	808	1.86 × 10^11^	0.97	0.34 s 0.67 s	[49]
2021	PbSe CQDs/CsPbBr_3_	365	8.97 × 10^12^	7.17	0.5 ms 0.78 ms	[53]
2022	PbSe CQDs/MoS_2_	635/808	3.17 × 10^10^@635 nm 2.65 × 10^10^@808 nm	23.5@635 nm 19.7@808 nm	0.36s@635 nm; 0.38s@808 nm 0.52s@635 nm; 0.86s@808 nm	[52]
2022	PbSe CQDs/CsPbBr_1.5_I_1.5_	532	5.96 × 10^13^	6.16	350 ms 375 ms	[54]
2022	PbS CQDs	940	2.1 × 10^12^	--	1.15 μs 0.49 μs	[19]

## Data Availability

Not applicable.

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
