# Peer review of "Lead Chalcogenide Colloidal Quantum Dots for Infrared Photodetectors"

_materials, 2023, doi:10.3390/ma16175790_

Round 1
Reviewer 1 Report
1. This review lacks essential information on lead chalcogenides, including their structure, properties, and other relevant aspects.
2. The author ought to include a dedicated section discussing the outlook and future perspectives of the related topic.
3. The author should incorporate a section that explores the advantages of colloidal quantum dots (CQDs) for infrared photodetectors in comparison to other forms of lead chalcogenides.
Minor editing of English language required.
Reviewer 2 Report
n this review article, the authors provided an interesting summary on the lead chalcogenide-based quantum dot for photodetectors. Authors have summarized the fabrication methods, as well as discussed the device applications of the lead chalcogenide quantum dots. The article will serve as a good reference for the scientific communities working on the lead-based IR and broadband photodetectors. Reviewer recommends the publication of the article with the following suggestions.
· Authors have highlighted mainly fabrication and device structures in the figures, putting more of the device characteristics or response curves like fig. 2 or fig. 4 would be more helpful to the readers to directly detect the performance improvement, so it is recommended.
· The article has also included hybrid materials and broadband photodetectors eg. 2d materials/quantum dots, please highlight in the introduction and modify it appropriately.
· Please check and correct figure references eg. Line 407
· Please check the spellings and sentences, eg. Line 92 and 114
Please check spellings and the sentences for eg. line 92, "was significantly "
